# Porosity evolution of mafic crystal mush during reactive flow

Matthew L. M. Gleeson[1,2] ✉, C. Johan Lissenberg[1] & Paula M. Antoshechkina [3]

The emergence of the "mush paradigm" has raised several questions for conventional models of magma storage and extraction: how are melts extracted to form eruptible liquid-rich domains? What mechanism controls melt transport in mush-rich systems? Recently, reactive flow has been proposed as a major contributing factor in the formation of high porosity, melt-rich regions. Yet, owing to the absence of accurate geochemical simulations, the influence of reactive flow on the porosity of natural mush systems remains under-constrained. Here, we use a thermodynamically constrained model of melt-mush reaction to simulate the chemical, mineralogical, and physical consequences of reactive flow in a multi-component mush system. Our results demonstrate that reactive flow within troctolitic to gabbroic mushes can drive large changes in mush porosity. For example, primitive magma recharge causes an increase in the system porosity and could trigger melt channelization or mush destabilization, aiding rapid melt transfer through low-porosity mush reservoirs.

For most of the last century, models of igneous differentiation and crustal magma transport have been dominated by the hypothesised presence of large, melt-rich magma chambers[1–4]. Within the last few decades, however, there has been a paradigm shift in how magmatic systems are pictured, moving away from the magma chamber hypothesis and towards a model in which magma reservoirs are dominated by crystal-rich mush[5–10].

The importance of mush-dominated magma reservoirs was first recognised at mid-ocean ridges, where geophysical surveys failed to identify the presence of large, melt-rich magma chambers[11,12], and petrological analyses revealed chemical variations that cannot easily be explained by crystallisation in large, well-mixed magma chambers[13–15]. Petrological and geochemical observations of erupted volcanic products and their cumulate counterparts, from both ocean island and arc volcanic systems, have since confirmed the importance of mush-dominated storage regions worldwide[16–22]. The scarcity of geophysical evidence for melt-rich magmatic systems (i.e., >50% melt) in the upper crust beneath active volcanic regions continues to support the mush paradigm[23,24], although the limitations of seismic

tomography for imaging thin (<200 m thick) melt-rich bodies has recently been highlighted[25].

The mush paradigm raises fundamental questions about melt transport: if magma plumbing systems are dominated by crystal mush, how is melt transported through them? How does melt accumulate to form eruptible reservoirs? Models for melt extraction or accumulation in mush-dominated systems have traditionally focused on the role of melt buoyancy and crystal compaction, squeezing the more buoyant magma out of the crystalline mush[26–29]. Evidence for crystal compaction is found in the form of plastic deformation of crystals from various magmatic systems worldwide that are shown to have large, long-lived mush reservoirs[17,30]; however, it remains unclear whether magma extraction and the solidification of mush systems can be driven by compaction alone[31,32]. As a result, attention has turned towards the influence of reactive flow, where a melt percolating through a porous crystal framework, possibly aided by mush compaction, reacts with the surrounding crystals[10,22,30,33–36]. Here, 'reactive flow' is used as a general term to describe both reactive porous flow along grain boundaries and focused flow in melt channels generated by reactive processes.

[1]School of Earth and Environmental Sciences, Cardiff University, Main Building, Park Place, CF10 3AT Cardiff, UK. [2]Department of Earth and Planetary Science, University of California Berkeley, McCone Hall, Berkeley, CA, USA. [3]Division of Geological and Planetary Sciences, Caltech, Pasadena, CA 91125, USA. ✉e-mail: gleesonm@berkeley.edu

Existing numerical models that incorporate parameterisations for both the physical and chemical aspects of reactive flow suggest that this process can generate melt-rich domains, channels or layers within crustal mush systems. They also indicate that reactive flow may be the dominant mechanism of melt transport and melt extraction in mush-dominated magmatic reservoirs[33,37]. However, current numerical models of melt transport incorporate highly simplistic empirical chemical parameterisations involving only a single solid composition, which may not accurately recreate the complexity of natural, mineralogically diverse magmatic systems[33,38]. The influence of reactive flow on the chemical and physical signature in gabbroic, troctolitic, and wehrlitic mush systems has also been investigated through petrological analysis combined with assimilation-fractional crystallisation (AFC) calculations[39–41]. However, as many AFC calculations are not energy-constrained, they likely violate the conservation of energy principle that governs the behaviour of natural systems (e.g., through the use of thermodynamically infeasible assimilated to crystallised mass ratios—$M_a/M_c$). As a result, the feasibility of reactive flow as a melt transport mechanism remains to be established, and our understanding of its influence on melt accumulation remains limited.

Here, we examine the influence of reactive flow on the porosity of troctolitic to gabbroic mush systems, relevant to mid-ocean ridge magmatism, using thermodynamic simulations implemented through alphaMELTS for MATLAB[42] (rhyolite-MELTS v1.0.2) that capture the full chemical complexity and phase equilibria of natural systems. Our models build on recent thermodynamic simulations performed using the Magma Chamber Simulator[43] by using a Bayesian approach to examine the influence of key input parameters (e.g., temperature, initial porosity) on the porosity evolution of mafic mushes. This approach also allows us to examine the role of kinetic factors during melt–mush reactions, such as variable mineral dissolution rates[44,45], which might influence the mineralogy of the reacted solid assemblage. Our model results reveal that the influence of reactive flow on the porosity evolution of mafic mush systems beneath active spreading centres is dominantly controlled by the phase proportions in the reacted solid material. In addition, our models highlight key conditions that might drive an increase in the system porosity, triggering mush fluidisation or the formation of melt channels, as well as situations where reactive flow causes a decrease in the system porosity, limiting future melt transport and likely driving the extreme chemical enrichment observed in magmatic cumulates[18,46,47]. The thermodynamic approach used here confirms that simple chemical parameterisations used in recent numerical models of reactive flow are insufficient to capture the complex behaviour of natural magmatic systems[33] and highlight that reactive porous flow could be key to melt transport within magmatic mush zones. Overall, our results have substantial implications for the construction of magmatic mush systems, melt transfer within high-viscosity, crystal-rich reservoirs, and the dynamics of magmatic systems in the build-up to volcanic eruptions.

## Results and discussion

### A thermodynamic approach to melt–mush reaction
Geochemical and textural signatures of reactive flow have been observed in mafic cumulate rocks from a wide range of tectonic settings worldwide, including mid-ocean ridge[40,46,48], ocean island[18], and arc volcanic systems[22]. However, current chemical and dynamical models of this process lack the ability to accurately determine how reactive flow in a heterogeneous crystal framework influences the porosity of a mush, which hampers our understanding of the implications of reactive flow for the efficiency of melt transport. To address this problem, we simulate the interaction of basaltic melts with olivine + plagioclase ± clinopyroxene mush systems in a series of dissolution–reprecipitation reactions using alphaMELTS for MATLAB (rhyolite-MELTS v1.0.2)[49,50]. The flexibility of our new model enables us to examine the influence of key parameters on mush porosity

evolution, including the composition and mineralogy of the initial crystal framework and the reacted assemblage, as well as the initial porosity and temperature of the system. Our work uses mid-ocean ridge magmatic systems as a template, as the influence of reactive porous flow has been well-documented in these systems, providing a wealth of data to compare to our models[36,40,41,46,47,51]. In addition, mid-ocean ridge magmatic systems are typically water-poor and hydrous minerals such as amphibole or biotite are rare, thus minimising errors resulting from the absence of accurate thermodynamic data in the MELTS models for these hydrous phases.

Initially, to determine whether our models provide an accurate simulation of natural mush processes, we create 200 reactive flow paths over a range of temperatures (1150–1230 °C) where a basaltic melt phase reacts with a gabbroic mush zone, and track key chemical parameters over six melt–mush reaction cycles. Results demonstrate that signatures linked to reactive flow in mid-ocean ridge settings are recreated by our models. These include the anomalously high $TiO_2$ signature and high Mg# (>90) of gabbroic clinopyroxenes, which cannot be generated by fractional crystallisation models alone, yet are frequently observed in the rock record[30,40,52] (Fig. 1). Consequently, our energy-constrained reactive flow models, which require far fewer assumptions than traditional assimilation-fractional crystallisation calculations[39,40], provide a meaningful simulation of the chemical reactions that occur in natural magmatic mush systems. As such, we can use these models to assess the influence of reactive flow on the porosity evolution of magmatic systems. To do so, we track the Melt Mass Ratio ($M_{Final}^{Melt}/M_{Initial}^{Melt}$; the ratio of the melt mass prior to, and after melt-mush reaction) of the local chemical system through the reaction.

We consider three scenarios that are appropriate to reactive flow within mid-ocean ridge magma reservoirs. First, we simulate the interaction of a cotectic, three-phase (olivine + plagioclase + clinopyroxene) saturated basaltic melt, which lies along the trend defined by fractional crystallisation of a mantle-derived parental magma, with a gabbroic mush ($Ol_5$:$Plag_{55}$:$Cpx_{40}$). A second scenario considers higher temperature simulations, where the initial melt phase is only saturated in olivine and plagioclase (i.e., undersaturated with respect to clinopyroxene), but all other parameters are equivalent to those used in scenario 1. Finally, we consider infiltration of 2-or-3 phase saturated basaltic melts into a troctolitic mush system ($Ol_{25}$:$Plag_{70}$:$Cpx_5$) over a range of temperatures (1150–1230 °C).

In each simulation, the initial porosity and temperature of the mush, as well as the mass of reacted, or assimilated, or assimilated solid material relative to the mass of melt (which can be used to calculate the melt/solid ratio) are randomly selected from a uniform distribution to test their influence on the model results (see Supplementary Information). In addition, by considering the mineralogy of the dissolved solid assemblage separately from the mineralogy of the mush system we can evaluate the roles of varying dissolution rates and mineral stability. Additionally, our different scenarios allow the importance of phase saturation in the liquid component to be evaluated.

There is abundant evidence in natural magmatic systems that reactive flow proceeds via dissolution–reprecipitation reactions as modelled here[41,46,48]. However, the exact parameters of these reactions, for example, how much dissolution occurs before precipitation is triggered, remain unclear. By considering this problem using a Bayesian approach to sample a broad parameter space for variables such as the mass of reacted material at each step, we can determine the sensitivity of reactive flow to these critical yet under-constrained parameters.

### Reactive controls on mush porosity
450 simulations were run for each of the three scenarios. These simulations reveal that during infiltration of three-phase saturated basalts into a gabbroic mush system (i.e., Scenario 1), the modal proportion of clinopyroxene relative to plagioclase increases in the mush,

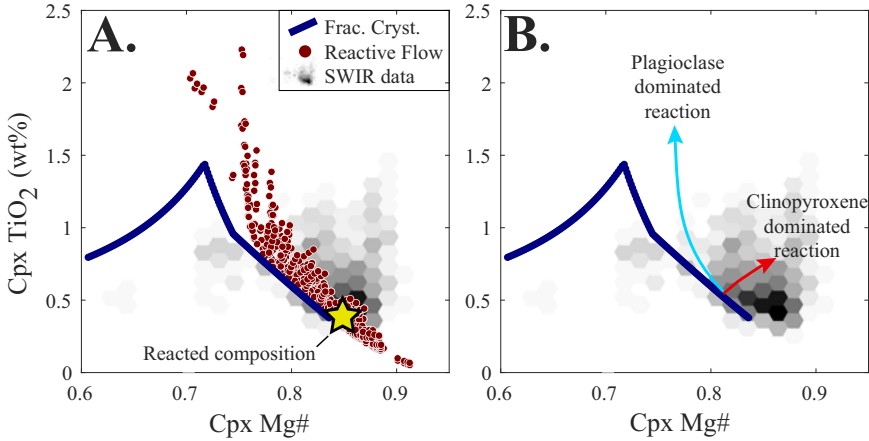

**Fig. 1 | Consequences of reactive flow on clinopyroxene compositions.** Fractional crystallisation models, using an initial melt composition based on primitive basalts from the Southwest Indian Ridge (SWIR; see the "Methods" section), predict clinopyroxene compositions that bracket the lower TiO₂ contents observed in natural mid-ocean ridge gabbros (data from ref. [40]). Reactive flow models were run over a range of temperatures (1150–1230 °C) with other parameters equivalent to those used in Scenarios 1 and 2 (see the "Methods" section and Supplementary Information). **A** TiO₂ enrichment is observed in many modelled clinopyroxenes, consistent with the natural data. In addition, reactive flow can also cause crystallisation of high Mg# (>0.90) clinopyroxenes−higher than any values observed in crystallisation models or experimental data[70], but in line with the natural data. **B** Schematic representation of the different paths followed by models dominated by plagioclase and clinopyroxene dissolution.

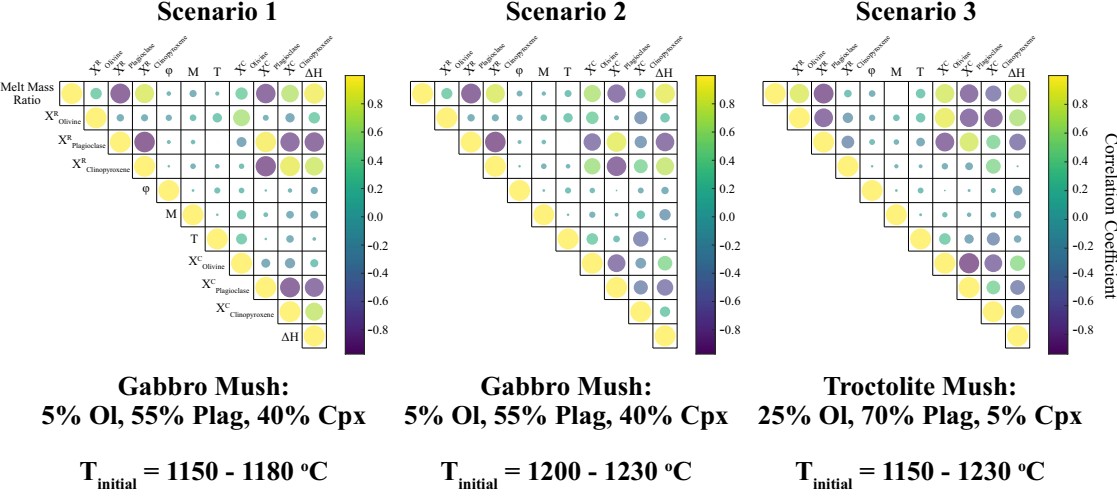

**Fig. 2 | Correlation matrices for each melt−mush reaction scenario.** In all models there is a strong correlation between the melt mass ratio, the proportion of plagioclase in the dissolved/reacted solid assemblage ($X^R_{Plagioclase}$), and the change of specific enthalpy in the melt phase (ΔH: defined as the difference between the specific enthalpy of the melt phase prior to, and after melt−mush reaction). There are, however, subtle differences between the scenarios. For example, in scenario 2, there is a weaker correlation between the proportion of clinopyroxene in the crystallised assemblage ($X^C_{Clinopyroxene}$) and the melt mass ratio than there is in scenario 1. This is likely due to the higher temperature of the scenario 2 models and the clinopyroxene-undersaturated nature of the initial melt phase (and thus the lower contribution of clinopyroxene to the crystallising assemblage).

consistent with reactions proposed based on observations of natural systems[30] and thus further supporting the validity of our models (see Supplementary Information). In fact, in all three scenarios many models with low proportions of plagioclase in the reacted solid assemblage ($X^{Plagiocase}_{reaction} < 0.2$) predict that the dissolved solid material shifts the local chemical systems far enough from the olivine−plagioclase−clinopyroxene cotectic that no plagioclase crystallisation is expected (Supplementary Information). Nevertheless, there is generally a strong correlation between the proportion of a given mineral in the reacted solid assemblage and the proportion of that phase in the crystallised component (Fig. 2).

We further note that, for the parameters chosen in this study, the melt mass ratio of reaction is typically centred around 0.95−1 (Fig. 3). In fact, when the randomly selected dissolved solid assemblage (see the "Methods" section) approximates the mineralogy of the mush

system, the melt mass ratio is consistently between 0.9 and 1.05. As a result, unless other factors drive the preferential dissolution of one phase relative to the others, which could occur due to differing dissolution rates and/or variations in the melt saturation state, reactive flow often has little to no influence on the porosity of mush systems: dissolution and reprecipitation are close to equal, and reactive flow proceeds by a process akin to zone refining[53,54]. While this result is dependent on other variables, as explored in more detail below, it indicates that reactive flow may operate 'freely', continually working to transport melt through magmatic mush systems and enabling chemical modification of the melt and crystal phases due to the ongoing reactions. However, while the mean melt mass ratio in all scenarios is ~1, the total range of values observed extends from ~0.7 to ~1.3, indicating that under certain conditions, reactive flow can have substantial impacts on mush porosity and thus the efficiency of melt transport.

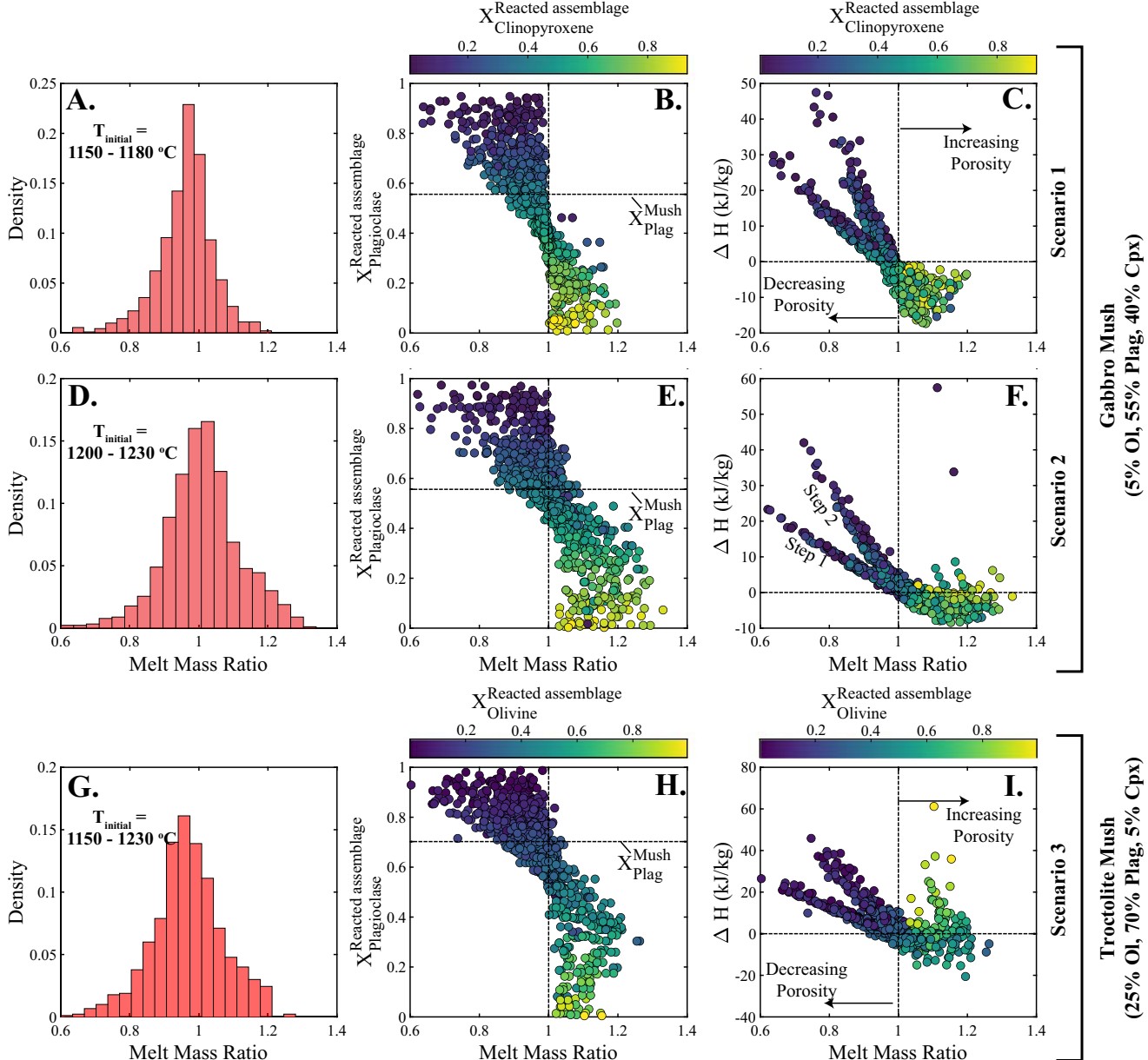

**Fig. 3 | Reactive flow controls on the system porosity.** The change in the melt mass through a reaction is described as $\frac{Mass_{Final}^{melt}}{Mass_{Initial}^{melt}}$ (melt mass ratio), and is compared to the proportion of plagioclase in the reacted solid assemblage ($X_{Plagioclase}^{Reacted\ assemblage}$) and the change in the specific enthalpy of the melt phase ($\Delta H$). Each simulation (450 per scenario) was allowed to run for 2 steps of melt–mush reaction. **A–C** Scenario 1: melt–mush reaction simulations involving an ol + plag + cpx saturated basaltic melt. Results indicate that the melt mass ratio during reactions tends to be clustered around 0.95–1 (**A**). There is a strong negative correlation between the melt mass ratio and the proportion of plagioclase in the assimilated assemblage (**B**), likely associated with the change in the specific enthalpy of the melt phase (**C**). **D–F** Scenario 2: high-temperature models, otherwise equivalent to scenario 1, where the initial melt phase is saturated in olivine and plagioclase. All results (for a given reacted mineralogy) are shifted to slightly higher melt mass ratios then in Scenario 1. **G–I** Scenario 3: the mineralogy of the mush system represents a troctolite, rather than a gabbro. The melt mass ratio increases with decreasing proportions of plagioclase in the reacted assemblage. All models were run at 100 MPa.

Consideration of all simulations in each scenario reveals that variations in mush temperature, initial porosity, and the solid/melt ratio have little influence on the melt mass ratio of reaction (Fig. 2). Instead, our simulations indicate that the mineralogy of the dissolved solid assemblage, here treated independently from the mineralogy of the mush to account for kinetic factors such as variations in mineral dissolution rates[55], represents the dominant control on porosity changes within mush systems. In natural systems, the temperature and melt composition may influence the mineralogy of the reacted assemblage contributing to the reactions (as discussed below), but these factors are poorly constrained and thus not incorporated into our models.

All scenarios reveal a clear correlation between the melt mass ratio and the proportion of plagioclase in the reacted assemblage ($X_{reaction}^{Plagiocase}$). This correlation is driven by the large latent heat component of plagioclase (relative to olivine and clinopyroxene)[56], and the change in the specific enthalpy of the melt phase during plagioclase-dominated reactions (Fig. 3). Specifically, when $X_{reaction}^{Plagiocase} > 0.5 - 0.7$, the positive change in the specific enthalpy of the melt phase requires excess crystallisation to balance the enthalpy of the system, and thus drives a decrease in the system porosity. When $X_{reaction}^{Plagiocase} < 0.4 - 0.6$, however, the melt mass ratio is often ~1 or >1, indicating that the porosity is either constant, maintaining the flux of melt through the system, or increasing, enhancing melt transport (Fig. 3).

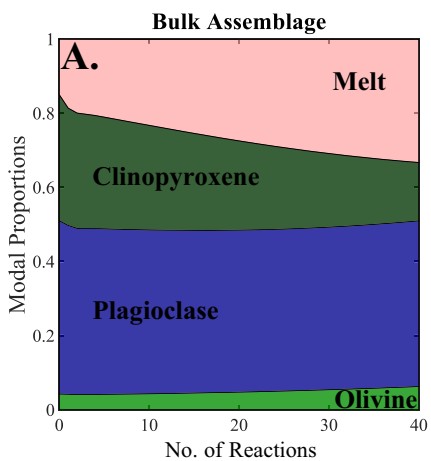

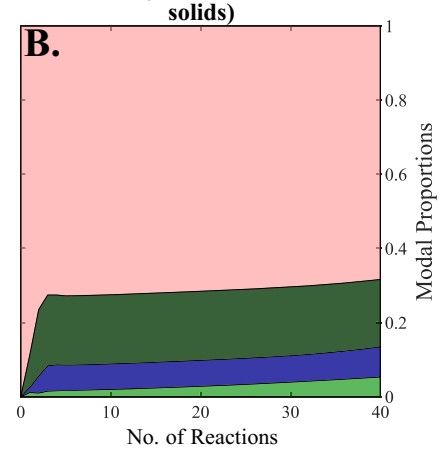

**Fig. 4 | Results of melt−mush reaction caused by repeated infiltration of clinopyroxene-undersaturated melts. A** Phase proportions in the entire mush system (i.e., melt, reacted solids and unreacted solid assemblage). **B** Phase proportions excluding solid material that remains chemically isolated from the melt phase (i.e., unreacted solid material). In essence, this panel represents the phase proportions of the reaction at each step of the model. Prior to any model reactions (step 0), the local chemical subsystem is composed entirely of melt. The clinopyroxene-undersaturated nature of the melt phase entering the mush system is assumed to drive clinopyroxene dissolution; as a result, clinopyroxene forms the dominant component within the reacted solid component (0.7) and the melt fraction of the mush increases with each reaction. If this process were to continue, it may be expected that the decrease in crystal fraction of the mush could eventually lead to partial remobilisation or channelisation within the mush system.

In addition, offsets between scenario 1 and 2 simulations reveal that the phase saturation of the melt component (i.e., the mineral phases that are stable at the liquidus) also plays an important role in the porosity evolution of a mush. Specifically, our models show that the interaction of olivine- and plagioclase-saturated, but clinopyroxene-undersaturated melts with a gabbroic mush system can cause an increase in the system porosity as long as $X_{\text{reaction}}^{\text{Plagiocase}} < 0.6$ (Scenario 2), whereas this transition occurs at -0.4 for melts saturated in a cotectic phase assemblage of olivine, plagioclase, and clinopyroxene (Scenario 1; Fig. 3). The offset between the two scenarios is related to the high latent heat component of plagioclase, which forms a larger proportion of the crystallising assemblage for models where the initial melt phase is clinopyroxene-undersaturated (Supplementary Information). Consequently, less crystallisation (of a plagioclase-rich assemblage) is required to balance the enthalpy of the system. Overall, these results indicate that melt transport through magmatic mush zones can be maintained, or even enhanced by reactive flow in situations where either the melt phase is clinopyroxene-undersaturated or plagioclase does not dominate the dissolved solid assemblage.

To investigate this further, we simulate the recharge of clinopyroxene-undersaturated melt into a gabbroic mush zone by modifying our model to consider multiple episodes of melt-mush reaction within a single volume of crystal mush. Results confirm that, if there is a sufficient flux of new melt into the mush to maintain clinopyroxene-undersaturation and/or clinopyroxene remains the dominant component in the reacted assemblage, this process will cause a dramatic increase in mush porosity (Fig. 4).

However, melt−mush reaction will not always maintain or enhance melt transport: in systems where $X_{\text{reaction}}^{\text{Plagiocase}} > 0.5 - 0.7$ reactive flow will cause a decrease in the porosity of the system. Critically, experimental analysis of mineral dissolution rates indicates that plagioclase dissolution occurs at a faster rate than that of olivine or clinopyroxene and that the activation enthalpy of dissolution for plagioclase is typically smaller than that of olivine[44,45,55], indicating that this situation may be favoured in many natural systems. Additionally, several petrological studies have shown that key trace element signatures of reactive flow, including the anomalously enriched compositions in clinopyroxene crystal rims[30,40], can be reproduced by models

where $X_{\text{reaction}}^{\text{Plagiocase}} > 0.5$. Therefore, we suggest that plagioclase-dominated reactions−that is, excess plagioclase dissolution relative to the modal proportions of plagioclase in the mush−might be the norm in cooling natural systems (outside of the influence of recharging melts, etc.). If so, the reactive flow of 3-phase saturated magmas within gabbroic mush systems could be partially responsible for the solidification of magmatic mushes, as well as the generation of enriched rims of low-An plagioclase and low-Mg# clinopyroxene beneath mid-ocean ridges, reducing the requirement of crystal compaction in generating cumulate rocks from cumulate mush.

## Implications for melt transport in natural systems

Compaction of crystal mush, which has long been postulated to be responsible for the formation of eruptible magma bodies and cumulate rocks[33,57], drives the porous flow of melts out of the initial mush to accommodate its compaction. Because melts are unlikely to be in equilibrium with the entirety of the solid matrix that they traverse, this porous flow may be reactive. As the melt mass ratio of most reactions is typically centred around 0.95−1 (Fig. 3), our findings indicate that porous flow might continue with little-to-no change in porosity and effectively contribute to the formation of melt-rich magma reservoirs and complementary cumulates within crystal-rich systems through compaction.

Evidence from natural systems, however, indicates that alternative mechanisms of melt transport are required beyond distributed porous flow. For example, in many fossilised mid-ocean ridge magma reservoirs, erupted lavas and sheeted dykes are in Fe−Mg equilibrium with deep, primitive regions of the mush, but not with the shallower, more evolved sections[10,58,59]. This observation necessitates the rapid and efficient transfer of melt through the magmatic mush to avoid magma re-equilibration with the more evolved, shallower gabbros. As such, melt transport by a distributed porous flow cannot explain these compositional relationships, indicating that channelised melt flow is required[10]. As dykes are rarely observed crosscutting MOR plutonic sections, this rapid transport must occur at supersolidus conditions, likely in the form of melt channels.

Melt channels generated by focused porous flow have been observed on a centimetre- to metre-scale in oceanic core complexes and ophiolites[41,48]. A reactive origin to these channels is supported by

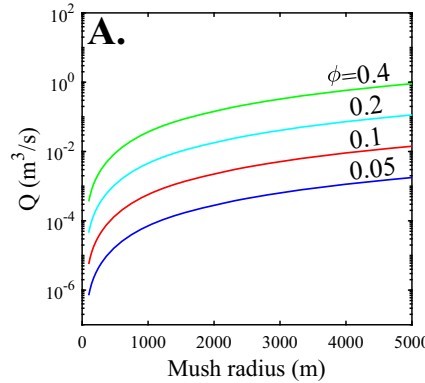
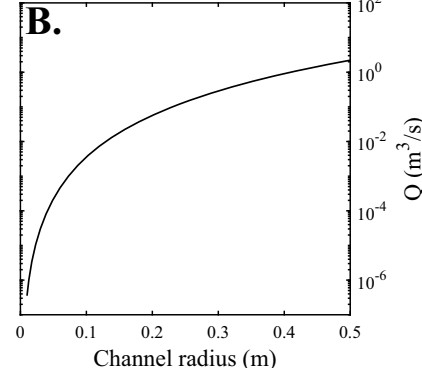

**Fig. 5 | Volumetric flux (Q) of magma through magmatic mush systems.** Comparison of volumetric flux calculations for porous flow (**A**) and channelised melt flow (**B**) demonstrate that channelised melt flow—with channel radii between 10 and 30 cm—can match the volumetric flux of melt through a crystal mush system 1–3 km in radius with a porosity (Φ) between 0.1 and 0.4. Volumetric flux calculations are performed using the equations displayed in the "Methods" section, with phase densities taken from the alphaMELTS for MATLAB fractional crystallisation model at 1180 °C and the mush density calculated using the modal proportions used in scenarios 1 and 2 ($ol_5$:$plag_{55}$:$cpx_{40}$). The viscosity of the melt phase is calculated in the Python3 tool Thermobar[67] using the melt viscosity model of Giordano et al.[68] at 1180 °C and the liquid composition from alphaMELTS for MATLAB.

their diffuse channel boundaries, mineral compositions indicative of reaction (e.g., clinopyroxene Ti–Cr relationships[30]), and the absence of a shape-preferred orientation in the surrounding crystal mush (indicating limited local deformation driving magmatic flow)[30,41]. Whether such channels also operate on larger scales, like their dunite counterparts in the mantle[60], remains to be determined. Nonetheless, simple back-of-the-envelope calculations confirm that buoyant melt flow within channels is capable, in principle, of transporting significant amounts of magma through magmatic mush systems. For example, the volumetric flux of a buoyant melt within a single 20 cm radius channel is equivalent to that estimated for distributed porous flow through a cylindrical mush system ~3 km in radius with 20% porosity (~0.05 m³/s; Fig. 5). Our models confirm that reactive flow can drive a dramatic increase in mush porosity and potentially contribute to the formation of these channels. Specifically, increases in mush porosity are likely when clinopyroxene is not on the liquidus and/or olivine or clinopyroxene dissolution occurs at a greater rate than that of plagioclase (Fig. 3). In troctolitic mush systems, olivine dissolution and removal during melt–mush reaction, alongside precipitation of secondary clinopyroxene in-channel structures, has been observed[48], and could indicate increasing mush porosity and the formation of melt channels during reactive flow. In gabbroic systems, clinopyroxene-undersaturated melts might drive increased dissolution of clinopyroxene, with both factors acting to increase the system porosity. Therefore, we suggest that an increase in gabbro mush porosity, and potentially the formation of melt channels, is likely during reactive flow at high temperatures, prior to clinopyroxene crystallisation (typically >1180–1200 °C for MORBs).

One scenario that promotes the interaction of a high-temperature, clinopyroxene-undersaturated magma with a magmatic mush is replenishment. In fact, our models indicate that flushing of primitive, high-temperature magmas—potentially originating from a recent sill intrusion—through a gabbroic mush could trigger a substantial increase in mush porosity (Fig. 4), a conclusion supported by recent thermodynamic calculations performed using the Magma Chamber Simulator[43,61]. In addition, clinopyroxene dissolution following magma replenishment has recently been documented using high-resolution chemical mapping and trace element analysis of large clinopyroxene oikocrysts in the Pacific lower oceanic crust[62]. Therefore, our models demonstrate that the dissolution of a clinopyroxene-bearing assemblage by a high-temperature, clinopyroxene-undersaturated replenishing melt, combined with the advective and conductive transfer of heat energy into the mush from the new replenishing melt, will drive an increase in the system porosity. We further suggest that rapid, focused melt extraction may be linked to these replenishment episodes as natural variations in the initial mush porosity or mineralogy could lead to spatial heterogeneity in the rate at which the mush porosity increases, focusing on melt flow, and the formation of reactive melt channels. Alternatively, dramatic increases in the system porosity surrounding a melt lens formed from a replenishing melt could contribute to mush destabilisation and trigger channelized melt transport (Fig. 6).

Once a channel is formed, our models indicate that reactive transport can proceed unimpeded if $X^{\text{Plagiocase}}_{\text{reaction}}$ remains low (<0.4 – 0.6). However, if $X^{\text{Plagiocase}}_{\text{reaction}}$ increases, the reactions will drive a decrease in the system porosity limiting future melt transport and possibly generating the extreme chemical enrichment observed in mineral rim zones[30,40,46].

### Broader implications
Overall, our results demonstrate that reactive flow can have substantial impacts on the porosity of magmatic mush systems, with implications for the efficiency of melt transport and the viscosity of magmatic reservoirs. This work also highlights that more detailed chemical interactions, which account for variations in the mineralogy of the mush and phase saturation of the melt, must be considered in larger, reservoir-scale models of reactive flow and melt transport to fully capture the dynamics of chemically heterogeneous and complex magmatic systems. At present, large-scale models that account for melt and crystal transport in km-scale magmatic systems contain only very simple chemical parameterisations, with all solid material described by a single composition at temperatures above the solidus[33,37]. Our results, however, demonstrate that the heterogeneous, multi-component nature of natural mush systems is critical for accurately modelling the influence of reactive flow and evolution of porosity within magmatic mush systems.

## Methods
### A thermodynamic solution to melt–mush reaction
We simulate the interaction of basaltic melts with mafic mush systems using the alphaMELTS for MATLAB package and the rhyolite-MELTS v1.0.2 thermodynamic model[49]. Our intention here is not to provide a complete physio-chemical model of melt and crystal transport in magmatic systems, but to construct a thermodynamic framework with which to evaluate the physical consequences of reactive flow.

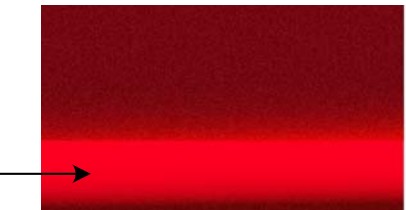

**(1) Replenishment. High-T, cpx-undersaturated magma intrudes a gabbroic mush.**

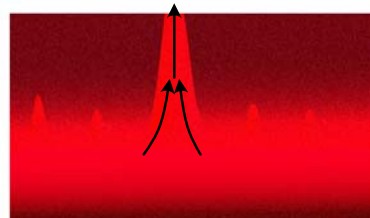

**(2) Reactive infiltration. Cpx-undersaturated magma ascends into overlying mush and drives an increase in porosity.**

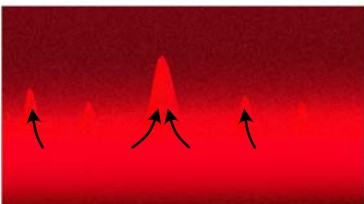

**(3) Melt focusing. Mush heterogeneity or melt flow instabilities cause focusing of melt flow. Porosity increase is greatest at certain locations.**

**(4) Channelised flow. Continued melt focusing and mush disaggregation may lead to channelised melt transport.**

**Fig. 6 | Schematic representation of the processes leading to melt channelisation.** The process proposed in this study initiates with a new intrusion of hot, primitive magma into a mafic mush system, likely in the form of a sill intrusion (**1**). The replenishing magma (**1**) infiltrates into the overlying mush and drives an increase in porosity due to the phase stability of the melt and the presence of clinopyroxene in the reacted solid assemblage (**2**). Heterogeneity in the mush system, or melt flow instabilities could lead to the focusing of melt flow into a specific region of the mush (**3**). Melt focusing could ultimately lead to the generation of a melt channel and rapid transfer of magmas within mush reservoirs (**4**).

We use our new code to develop two models of melt–mush reaction. The first considers the evolving nature of a melt phase as it passes through a magmatic system and chemically interacts with multiple regions of mush (Figs. 1 and 3); i.e., the melt composition formed in each reaction proceeds into a new region of uninfluenced mush where melt–mush reaction occurs again. The second model focuses on a single mush horizon and considers the influence of several episodes of melt–mush reaction (due to ongoing melt flow into the mush) on the mineralogy, composition, and porosity of that region (Fig. 4).

To ensure the results of these models are appropriate to the natural system, we must consider both the local chemical system within which the reaction occurs (melt + solid reactants) as well as the thermal properties of the wider mush system. Chemical mapping of natural cumulate samples reveals the presence of 'relict' crystal cores that appear to be uninfluenced by melt–mush reaction, indicating that complete chemical equilibrium across the mush system is rarely achieved[30,41]. As a result, we define two chemical sub-systems, one characterised by a melt phase and the solid material involved in the reaction (i.e., the dissolved assemblage), and the second characterised by an unreacted solid assemblage. However, as the thermal diffusivity of basaltic melts and mafic mineral phases is several orders of magnitude greater than the chemical diffusivities that control mineral dissolution rates[44,55,63], we assume that the local mush system (including both the unreacted solid assemblage and the active chemical system) remains in thermal equilibrium during melt–mush reaction. Therefore, to determine the new equilibrium assemblage for the mush system following melt–mush reaction we consider both the chemistry of the local subsystems and the energy balance across the mush (i.e., the unreacted solid assemblage may act to buffer any temperature variations caused by melt–mush reaction).

Our new models of reactive flow and melt–mush reaction require several key independent variables to be specified by the user, such as the temperature and pressure of the mush system prior to reactive flow ($T_{initial}$, $P$). Our model also offers the opportunity to provide a temperature offset between the mush and the percolating melt phase ($\Delta T$), but this is set at 0 for all calculations shown in this study. Other variables include the initial melt mass fraction of the mush system ($\varphi$; taken here to broadly represent the mush porosity) and the mass of solid material involved in the reaction relative to the mass of melt in the system ($M$; i.e., the mass of dissolved solid material), which can also be used to describe the melt/solid ratio of the reaction ($1/M$). These two variables are critical as they can be used to define the relative mass of the three key components within the system: solid material within the mush that remains isolated from reaction ($1-\varphi*(1+M)$), the mass of solid material that reacts with the percolating melt ($\varphi*M$), and the mass of melt ($\varphi$).

The composition of the initial melt phase and the solid phases in the mush system must also be specified ($C_{phase}$). In addition, it is also necessary to define the mineralogical proportions in the mush ($X_{phase}^{Mush}$) and the relative mineral proportions in the reacted assemblage ($X_{phase}^{React}$). Notably, we define these two variables separately, which allows us to investigate the influence of additional factors such as variations in dissolution rates (leading to excess incorporation of a single phase into the reaction). Once these variables have been defined, we can use the rhyolite-MELTS v1.0.2 thermodynamic model, implemented through alphaMELTS for MATLAB, to simulate reactive flow (Fig. 7)[49,50].

To simulate reactive flow, we first perform a simple fractional crystallisation calculation to convert the melt composition provided by the user into a related magmatic composition at the model-specified temperature. These fractional crystallisation models initiate at the liquidus of the user-provided melt phase and terminate at the temperature specified for the reactive flow models. At this point, the composition and specific enthalpy of the melt phase, critical for the following calculations, are extracted from the fractional crystallization models.

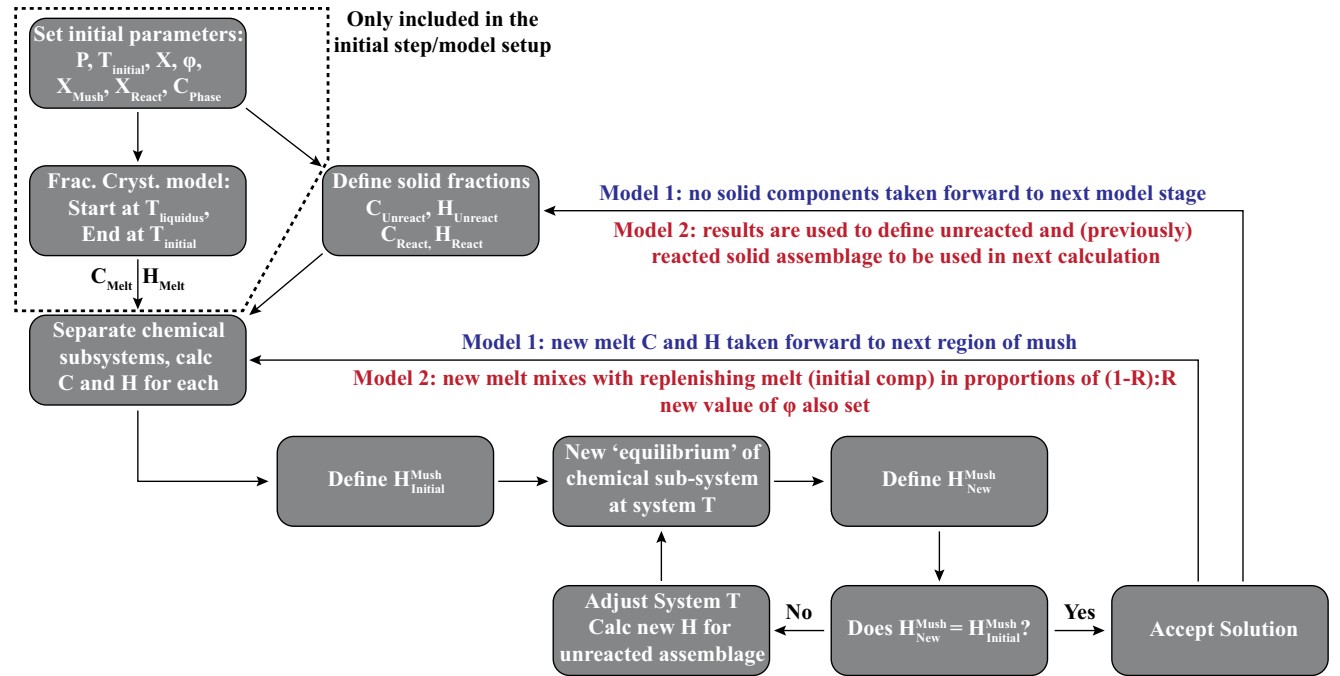

**Fig. 7 | Flow diagram outlining the two models developed in this study.** We provide two models to track the influence of melt-mush reaction on mush porosity. The first (Model 1) follows a single batch of melt as it reacts with multiple regions of magmatic mush. The second (Model 2) tracks the evolution of a single mush horizon as the new melt is continuously flushed through the system. C refers to the composition of the system (i.e., the concentration of any oxide of interest), and H refers to the specific enthalpy of the specified region.

Next, we use the specified values of $M$, $\varphi$, $X_{phase}^{Mush}$, $X_{phase}^{React}$, and the composition of each phase to create two chemical systems: the unreacted solid assemblage and the local chemical system of the reaction. In addition, once these chemical subsystems have been defined, we can use the '*calcPhaseProperties*' command in alphaMELTS for MATLAB to calculate the specific enthalpy for each phase in the solid mush; where appropriate, these calculations are performed separately for the solid material that is involved in the chemical reactions and the unreacted solid assemblage. Now that the specific enthalpy of each phase (and each component) within the mush system is known, the total enthalpy of the mush system can be calculated by

$$H^{Mush} = H^{liq} * \varphi + H^{Solid_{unreacted}} * (1 - \varphi*(1+M)) + H^{Solid_{reacted}} * M * \varphi \quad (1)$$

where $H$ represents the specific enthalpy of each component (J/kg). Within our models, we assume that there is no flux of heat energy into or out of the system during melt–mush reaction and, as these calculations are performed at constant pressure, this indicates that the value of $H^{Mush}$ must remain constant during the reaction (where $H^{Mush} = H^{Reaction} * \varphi * (1 + M) + H^{Solid_{unreacted}} * (1 - \varphi * (1 + M))$). Therefore, any change in the enthalpy of the local chemical system of the reaction ($H^{Reaction}$) must be balanced by changes in the enthalpy (and therefore temperature) of the unreacted solid assemblage ($H^{Solid_{unreacted}}$) and this constraint is critical for determining the new equilibrium assemblage within the mush.

In addition to the enthalpy of the mush system, we also need to consider the chemical composition of the local chemical system of the reaction. This is defined as a mixture of the melt phase and the reacted solid assemblage:

$$C^{Reaction} = C^{liq} * \frac{\varphi}{\varphi*(1+M)} + C^{Solid_{reacted}} * \frac{M*\varphi}{\varphi*(1+M)} \quad (2)$$

Once the chemical composition of the reaction is defined, we can proceed to calculate the mineralogical assemblage and phase

compositions in the local chemical system following melt–mush reaction. This is achieved using an iterative approach to ensure conservation of energy throughout the reaction (i.e., constant $H^{Mush}$). First, an initial 'guess' for the post-reaction assemblage is determined at the current temperature of the system and the enthalpy of this new assemblage is recorded ($H_{New}^{Reaction}$). As $H_{New}^{Reaction}$ is unlikely to match the enthalpy of the melt phase and reacted solid assemblage prior to the calculation ($H_{Initial}^{Reaction}$), this initial guess for the post-reaction assemblage will break the constraint that the internal energy of the system ($H^{Mush}$) must remain constant during the reaction. As a result, we iteratively adjust the temperature of the mush system, and recalculate the post-reaction local equilibrium (and $H_{New}^{Reaction}$) as well as the enthalpy of the unreacted solid assemblage ($H_{New}^{Solid_{unreacted}}$) at each temperature. This process is repeated until $H_{Initial}^{Mush} = H_{New}^{Mush}$, at which point the temperature of the system, alongside the new melt and mineral compositions–as well as their abundances–are recorded. At this point, the process may be repeated, either by taking the new melt composition forward to undergo reaction with a new region of magmatic mush (Model 1; Fig. 1) or by separating the solid fraction of the mush into a reacted and unreacted component, allowing the new melt to enter the same region of mush (where the relative fraction of new melt entering the system is defined by a parameter $R$) and a new reaction to proceed (Model 2; Fig. 4).

In theory, the model provided in this study is extremely flexible and can easily be expanded to consider other scenarios (e.g., the influence of reactive flow in silicic systems). Simplified MATLAB codes designed to simulate the interaction of basaltic melts with gabbroic mush zones are provided in a GitHub repository and the current version of these models are archived on Zenodo. These models can be easily modified to incorporate different mineral phases, melt compositions, or internal conditions (e.g., temperature and pressure). For the purposes of this study, however, our models are used to focus on the influence of reactive flow in mid-ocean ridge settings. As a result, in all models provided here, we use natural data from the South-West Indian Ridge and the nearby Atlantis Bank oceanic core complex to constrain the composition of the melt and mineral phases in our reactive flow

models. Specifically, we use the mean composition of samples KNO0162-9-048-021GL, KNO0162-9-048-004GL and AII0107-6-056-028GL (3 of the most primitive basalt compositions from the SWIR) from -52–55 °S on the SWIR as the initial melt composition in our fractional crystallisation and reactive flow models (compositions taken from the compiled database of Gale et al.[64]). Similarly, the composition of the mineral phases used in the reactive flow models are taken from the work of Boulanger et al.[40] on the Atlantis Bank oceanic core complex; in detail, we use the average core composition of minerals measured in olivine-gabbros and troctolites at depths greater than 500 m. Notably, fractional crystallisation models initiated at our proposed starting composition are able to recreate the major compositional trends observed in the SWIR basalts, with reactive flow potentially acting to explain some of the compositional spread around this primary differentiation path (see Supplementary Information).

Within the simulations shown in Figs. 1 and 3, we used a Bayesian approach to assess the influence of each parameter on the chemical and physical consequences of reactive porous flow. For all scenarios, the initial values of $T_{initial}$, $M$, and $\varphi$ in each simulation were selected from uniform distributions ($0.05 < M < 0.35$; $0.1 < \varphi < 0.35$; $T$ range specified separately for each scenario). The mineralogy of the mush is set to a constant value within each scenario, but the mineralogy of the dissolved solid assemblage is allowed to vary. This is done by creating a random value for each mineral phase ranging from 0 to their modal proportion in the wider mush system and then normalising the outputted values. For example, in Scenario 1, the code generates random values for olivine, plagioclase, and clinopyroxene between 0 and 0.05, 0.55, and 0.40, respectively. In a single simulation, the code may return hypothetical values of 0.0125, 0.4236, and 0.0890, which when normalised equates to modal proportions in the dissolved solid assemblage of 0.0238, 0.8067, and 0.1695 for olivine, plagioclase, and clinopyroxene, respectively. Which minerals are dissolved by percolating melts in natural systems, and their relative proportions, will likely be influenced by several factors, including the temperature of the system, the relative geochemical enrichment of the percolating melt relative to that which formed in the initial mush (especially as magmas of the lower crust likely retain much of the underlying mantle heterogeneity), and the kinetic barriers to dissolution/reprecipitation. As these factors are difficult to incorporate into our thermodynamic-based models, we believe our Bayesian approach, sampling a wide distribution space, is the most effective means of determining the influence of melt-mush reaction on mush porosity evolution.

### Volumetric flux calculations

We provide calculations for the volumetric flux of magma through a mush zone in two different flow regimes. First, we estimate the volumetric flux of magma that can be achieved by buoyancy-driven flow through a high-porosity melt channel. Second, we consider the volumetric flux of magma that results from buoyancy- and compaction-driven flow out of a cumulate mush zone. Comparing the two values provides insights into the role that melt channelisation and mush permeability play in the extraction of magma from magmatic mush zones and/or the transfer of magmas within a mush.

The volumetric flux of buoyant magma through a high porosity melt channel within a deformable mush zone can be calculated via[65,66]:

$$Q_C = \frac{\pi}{8} \times \frac{\triangle \rho g r^4}{\mu_l} \qquad (3)$$

where $Q_C$ is the volumetric flux of magma through the channel (m³/s); $\Delta\rho$ is the density difference between the mush and the ascending melt phase ($\rho_{melt}$-$\rho_{mush}$; kg/m³); $g$ represents gravitational acceleration (−9.81 m/s²); $r$ is the radius of the channel (m); and $\mu_l$ is the viscosity of the melt phase (Pa s). This equation, relating $\Delta\rho$ and $r$ to the volumetric magma flux, is taken from a simplification of plume flow[66], but

provides meaningful insights into the volumetric flux of magma that can be achieved in a melt channel if the viscosity of the melt is substantially lower than the medium it is passing through. As the mush has a viscosity several orders of magnitude greater than the melt phase, this condition is met. The calculations are performed using a range of different channel radii, other parameters are taken from alphaMELTS for MATLAB (that is, the density of the different phases) with the exception of the melt viscosity[67,68] (Fig. 5).

In contrast, the volumetric flux of magma through a permeable mush reservoir can be calculated via Darcy's Law[69]:

$$Q_D = \frac{kA}{\mu_l} \times \frac{\triangle P}{\triangle z} \qquad (4)$$

where $k$ represents the permeability of the mush zone (m²); A is the surface area of the mush zone (m²); and $\Delta P$ is the pressure gradient within the mush over vertical height $\Delta z$ (Pa/m). The pressure gradient generated by density differences between the melt phase and crystal framework within a mush zone can be written as

$$\triangle z = \frac{\triangle P}{\triangle \rho g} \qquad (5)$$

Therefore, by rearranging the above equations, the volumetric flux of magma through a permeable mush zone can be written as

$$Q_D = \frac{kA}{\mu_l} \times \triangle \rho g \qquad (6)$$

where the permeability of the mush zone is calculated using the method outlined by ref. 37:

$$k = a^2 b \phi^\alpha \qquad (7)$$

Here $a$ represents the typical radius of the grains within the mush zone ($m$); $b$ is a permeability constant (set to 0.002); $\alpha$ represents a permeability exponent (set to 3); and $\Phi$ represents the porosity of the mush.

## Data availability

This study produces no new data. All data used in the modelling can be found in the archived Zenodo repository associated with this study (https://doi.org/10.5281/zenodo.7626521).

## Code availability

All code developed in this study is available via the lead author's GitHub (https://github.com/gleesonm1/MeltMushRxn) and the current version used in this publication has been archived using Zenodo (https://doi.org/10.5281/zenodo.7626521). The GitHub repository is set up with a Results folder containing MATLAB scripts that will recreate the figures shown in this manuscript, allowing readers to investigate the results of our models in detail.

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

## Acknowledgements

This work was supported by a Research Fellowship awarded to M.L.M.G. by the Royal Commission for the Exhibition of 1851. C.J.L.'s work on reactive flow was supported by NERC grant NE/I001670/1. P.M.A. was supported by NSF grant EAR-1947616.

## Author contributions

M.L.M.G. and C.J.L. designed the study and came up with the initial framework/idea for the models. New code was written by M.G. who also performed all model calculations with input/help from P.M.A. The underlying alphaMELTS for MATLAB package used in this study was developed by P.M.A. Manuscript writing and figure construction were carried out by M.L.M.G. in consultation with C.J.L. who provided detailed edits and comments throughout the process. P.M.A. also contributed to the editing of the manuscript.

## Competing interests

The authors declare no competing interests.
