## [Peer Review File · Nature Communications]

Reviewers' Comments:

Reviewer #1:

Remarks to the Author:

Magma transport and porosity evolution in crystal mush systems

Reviewer: Valentin Basch

The article entitled Magma transport and porosity evolution in crystal mush systems, submitted to Nature Communications by Matthew Gleeson and co-authors, presents a comprehensive numerical modeling of reactive porous flow processes, constraining the impact of the assimilated material, melt composition and temperature on the evolution of porosity within a reacted gabbroic crystal mush.

This contribution is in line with topics of interest for Nature Communications and will very likely be of great interest to the community of igneous petrology. The manuscript is well written, organised and illustrated, and easy to follow throughout the text. The Supplementary Information and Methods are a useful (and needed) addition to the detailed understanding of the numerical models performed. I reckon that this study is a fundamental addition in the way reactive systems are described. Dynamic, evolving systems are now the rule and this contribution on Nature Communications allows to constrain the evolution of these complex environments. Numerical models are often expected/requested and providing a well-constrained tool to study the evolution of crystal mush systems is very welcome. Indeed, MELTS models are mostly melt-oriented and are often difficult to link with complex matrix-driven mush processes. The implication of the study on the formation of high-porosity melt channels during replenishment is also of great interest in constraining the flux of melt through crustal systems. I thereby recommend publication of this manuscript after minor revisions, detailed in the following.

Below, I list 4 main comments, before giving minor comments referred to the line numbers of the submitted PDF file.

Thank you for this promising contribution in the field of reactive melt transport.

Valentin Basch

Main comments:

1. Details of the models: One of the main conclusion of the manuscript is that during reactive migration, the assimilated assemblage governs the evolution of porosity of the reacted crystal mush. Throughout the text, the authors describe well the different parameters that are involved in the models, i.e., the initial mass fraction of the mush system (ϕ), the mass of reacted solid material (M), proportion of assimilated phases (XReact), but do not explicitly present the assimilated mass at given steps of decreasing temperature. I do understand that the Melt Mass Ratio best describes the evolution of the system, however, the assimilated mass (during one single evolution step) drives the variation in melt chemistry and therefore governs melt saturation and fractionated phases. This strongly impacts the specific enthalpy of the melt phase and therefore the excess crystallization needed to balance the system in the case of plagioclase-dominated reactions (Lines 136-144). These details of the model should be implemented and discussed.

One problem that concerns me when dealing with "realistic" crystal mush models is that, if I am not mistaken, the assimilated mass is fixed by the user before running the models. The assimilation step simply consists of the addition of the chosen components within the melt phase, before the thermodynamic software looks for reequilibration of the enthalpy of the system through crystallization of new phases (e.g., Lines 139-158). My question is: Would a natural system overshoot like this and drive excess crystallization, or would it rather not be able to dissolve? Although Liang (2003) did demonstrate that excess dissolution is common and related dissolution-recrystallization is ubiquitous, I think that a discussion of these processes and how they proceed in nature would be appreciated.

2. Starting melt composition: The authors do not justify the choice of averaging 3 specific samples to

obtain their starting composition for the models. Why not start from the primitive SWIR MORB estimated from Coogan et al. (2004) at the Atlantis Bank, for example? The starting melt composition governs the phase saturation and might result in very different results if clinopyroxene appears before or after on the liquidus. As the authors discuss the impact of the reacted matrix on the evolution of porosity, I think that they should thoroughly discuss the impact of the starting melt composition as well (Lines 511-521).

3. Formation of porosity channels: Throughout the text, the authors mention that dissolution of plagioclase occurs at faster rates than that of olivine and clinopyroxene (Lines 165-167). Additionally, the modal composition of the considered reacted mushes is dominated by plagioclase, from troctolitic (ol:plg:cpx = ~25:70:5 vol%) to gabbroic crystal mushes (ol:plg:cpx = ~5:55:40 vol%). It thus seems fair to assume that systems where plagioclase is dissolved at rates higher than $X_{\text{React}}(\text{plg}) = 0.5-0.6$ would be the most common in nature. How do the authors reconcile the preferential assimilation of plagioclase and the large quantity of plagioclase in gabbroic crystal mushes with the decrease in porosity in reactive systems dominated by plagioclase dissolution? Although in some cases, replenishment of clinopyroxene-undersaturated, high-temperature melt could indeed lead to increasing porosity of the system, shouldn't the porosity of the system be driven down as soon as clinopyroxene appears on the liquidus? Is there a way of creating porosity channels from more evolved, clinopyroxene-saturated melt? This has very strong implications since it preferentially links the most primitive lower crustal systems directly to the eruptible system.

4. Supplementary Figures 4-5-6: In the histograms of crystallized mass, I do not understand the significance of the peak in models that crystallized no plagioclase? Almost all models include plagioclase assimilation and the melt is initially plagioclase-saturated, it is hard to believe that so many models did not crystallize any plagioclase (Fig. 4H, 5H, 6H). I can understand that in a troctolitic world, models do not allow crystallization of cpx at high temperatures (Fig. 5I), but it is hard to believe for plagioclase, which is strongly assimilated and clearly the dominant phase on the liquidus (Fig. 6H). Same goes for all the models that crystallize 100 vol% olivine (Fig. 6G) in a high-temperature system saturated in plagioclase.

Minor comments:

Line 11-12: This is only one of these "several questions" raised, you should rephrase this sentence.

Line 16: "under-constrained"

Lines 38-43: Compaction and reactive flow are not mutually exclusive, as the former can very well drive the latter. During compaction of any crystal mush, extraction of the interstitial melt forces percolation within a crystal matrix that is likely not in equilibrium with this migrating melt. This chemical disequilibrium can trigger dissolution-precipitation processes along the way. It is true, however, that compaction processes are hard to properly constrain and have been only recently object of detailed EBSD studies (e.g., Bertolotti et al., 2019 *Geology*; Ferrando et al. 2021 *Tectonophysics*). The way the sentence is written opposes compaction to reactive flow, whereas they are two processes working together within a single environment.

Lines 45-46: By flow, do you intend both diffuse and focused percolation?

Lines 62-64: This information already comes in handy for decades of AFC models that used ratios of assimilated over crystallized mass ($r = M_a/M_c$) of 0.95-0.99, without any constrain whether this is feasible and realistic!

Line 94: "These include the anomalously high"

Lines 132-135: But the dissolution of these phases are function of the temperature of the system and

chemical composition of melt and crystal matrix. I think that this should be mentioned.

Lines 147-148: "interaction of olivine- and plagioclase-saturated, but clinopyroxene-undersaturated melts"

Line 178: compaction "drives" porous flow of melts, more than requires.

Line 188: "transfer of melt from through the"

Lines 193-194: "on a centimetre- to metre-scale in oceanic core complexes and ophiolites"

Line 196: "the absence of a shape-preferred orientation"

Lines 194-197: By the lack of SPO, do you mean that no extensive compaction was involved? I think that you should mention it directly, to avoid misunderstandings.

Line 390: "higher Melt Mass Ratios than in Scenario 1"

Figure 3: Please be more explicit in the figure caption, by dividing A and B, and explaining in detail why the melt phase decreases in proportion in B during the cpx-dominated reaction leading to an increase in porosity.

Line 448: Is the "initial melt mass fraction of the mush system" to be considered as the porosity of the system?

Line 464: "These fractional crystallisation"

Lines 522-533: This approach is used to capture the full variability of how the system can vary upon reaction. Is this likely that the system can lead to extensive dissolution of clinopyroxene, while plagioclase is left unreacted? A consideration of the equilibrium between the melt and the reacted crystal matrix could be used to filter the most extreme cases. I understand that the large database created this way spans all end-members of reaction and therefore gives the whole picture of the reaction-driven chemical variability; it must then be carefully analyzed and interpreted by the user.

Line 536: "buoyancy-driven"

Line 537: "high-porosity melt channel"

Line 538: "results from buoyancy- and compaction-driven flow"

Reviewer #2:

Remarks to the Author:

Dear Authors,

I read your manuscript with interest, although I feel ill-equipped to dig very deep on the more petrological aspects of the study.

This study is concerned with the role of reactive transport in maintaining permeable flow through mafic mushes and explaining changes in mineral assemblages associated with it. To this effect a significant number (>400) thermodynamic simulations with MELTS are run varying parameters such as the initial state of the mush (temperature, assemblage, starting porosity, melt composition) to obtain equilibrated new assemblages and porosity. I do believe that the present study fills an important gap, probing if the reactions that occur along flow paths are likely to impede or facilitate further transport as they progress. I think the manuscript makes a good case that for mafic systems

reactive porous flow is likely to maintain flow (no significant changes in porosity).

There are three easy to address issues I found with the manuscript and I will start from the simplest one and finish with perhaps the most tricky one.

(1) The text up to line 70+ is fairly general with respect to the types of magmatic systems considered. I did not realize before line 70 or so that this study was exclusively focused on mafic systems with mostly MOR considerations. A large fraction of the intro references papers that are written with more silicic and continental crust oriented systems. I would encourage the authors to make the text more specific from the start (even in the title) and reduce the intro to the discussion of mafic systems, as I have doubts that the present study provides significant evidences for the consequences of reactive flow in silicic mushes.

(2) The intro was a bit disappointing to read and, I am sorry to say, felt a bit lazy. First the lack of focus (see previous point) and second the choice of references used in the text is frustrating. About the latter, the authors rightfully point that the mush paradigm has been developed about 2 decades ago, yet all the references in the first paragraphs are recent and mostly from a single, yet prominent, group in a much larger community. Please cite the original studies that presented the mush paradigm as well as the recent studies that banked on it (see papers from Wes Hildreth, Olivier Bachmann and other papers by Bruce Marsh). Also update the references (line 32-33) about the geophysical support for the lack of melt-rich reservoirs with recent papers that explain that the limitations of seismic tomography are significant and their absence in the geophysical record not preclude their existence (see for example Rashtbehesht et al, JGR, 2020).

(3) It seems that most (all?) MELTS simulations were run under fixed enthalpy conditions. If this is correct, then the authors should clearly justify this choice and perhaps discuss some of the implications. My understanding (but I could be convinced otherwise) is that because of the equilibrium assumption built in these thermodynamic models, it is assumed that the final assemblage is fully reacted/equilibrated. How good of an assumption is that? Is it possible to consider that kinetically limited conditions can lead to different evolution paths and perhaps more porosity change than the comparison between initial and fully reacted assemblages (given that the small change in porosity is mostly attributed by the dissolution of certain minerals being balance by precipitation of other phases)? Under several instances, the text refers to melt transport, compaction processes or even rapid changes (line 215). These are misleading because the thermodynamic results presented here do not have lengthscales or timescales attached to them. I would be more careful with these statements. I would like to encourage the authors to discuss equilibrium and the absence of kinetics a bit more, especially in the discussion section. There is only so much one can do with MELTS type simulations.

Reviewer #1 (Remarks to the Author):

Magma transport and porosity evolution in crystal mush systems

Reviewer: Valentin Basch

The article entitled Magma transport and porosity evolution in crystal mush systems, submitted to Nature Communications by Matthew Gleeson and co-authors, presents a comprehensive numerical modeling of reactive porous flow processes, constraining the impact of the assimilated material, melt composition and temperature on the evolution of porosity within a reacted gabbroic crystal mush.

This contribution is in line with topics of interest for Nature Communications and will very likely be of great interest to the community of igneous petrology. The manuscript is well written, organised and illustrated, and easy to follow throughout the text. The Supplementary Information and Methods are a useful (and needed) addition to the detailed understanding of the numerical models performed.

I reckon that this study is a fundamental addition in the way reactive systems are described. Dynamic, evolving systems are now the rule and this contribution on Nature Communications allows to constrain the evolution of these complex environments. Numerical models are often expected/requested and providing a well-constrained tool to study the evolution of crystal mush systems is very welcome. Indeed, MELTS models are mostly melt-oriented and are often difficult to link with complex matrix-driven mush processes. The implication of the study on the formation of high-porosity melt channels during replenishment is also of great interest in constraining the flux of melt through crustal systems. I thereby recommend publication of this manuscript after minor revisions, detailed in the following.

Below, I list 4 main comments, before giving minor comments referred to the line numbers of the submitted PDF file.

Thank you for this promising contribution in the field of reactive melt transport.
Valentin Basch

Main comments:

1. Details of the models: One of the main conclusion of the manuscript is that during reactive migration, the assimilated assemblage governs the evolution of porosity of the reacted crystal mush. Throughout the text, the authors describe well the different parameters that are involved in the models, i.e., the initial mass fraction of the mush system (ϕ), the mass of reacted solid material (M), proportion of assimilated phases (XReact), but do not explicitly present the assimilated mass at given steps of decreasing temperature. I do understand that the Melt Mass Ratio best describes the evolution of the system, however, the assimilated mass (during one single evolution step) drives the variation in melt chemistry and therefore governs melt saturation and fractionated phases. This strongly impacts the specific enthalpy of the melt phase and therefore the excess crystallization needed to balance the system in the case of plagioclase-dominated reactions (Lines 136-144). These details of the model should be implemented and discussed.

We believe that our terminology in the manuscript and methods section may have led to some confusion here. We did, in fact, thoroughly test the influence of varying the mass of assimilated material (treated as equivalent to the mass of reacted solid material) on the results of our models. These results are shown in the histograms in the supplementary information (Figs. S.4-6) as well as in Fig. 2 which has been added to the main text (originally in the supplement). To clarify our approach, we have added a clause to L130-134 of the main text specifying that we treated the mass

of reacted/assimilated material as an unknown and investigated the influence of this parameter on our model results.

One problem that concerns me when dealing with “realistic” crystal mush models is that, if I am not mistaken, the assimilated mass is fixed by the user before running the models. The assimilation step simply consists of the addition of the chosen components within the melt phase, before the thermodynamic software looks for reequilibration of the enthalpy of the system through crystallization of new phases (e.g., Lines 139-158). My question is: Would a natural system overshoot like this and drive excess crystallization, or would it rather not be able to dissolve? Although Liang (2003) did demonstrate that excess dissolution is common and related dissolution-recrystallization is ubiquitous, I think that a discussion of these processes and how they proceed in nature would be appreciated.

This is a very interesting comment and we have addressed it by including a brief discussion of how our models account for these under-constrained variables (e.g., the amount of dissolution ‘overshoot’ that takes place prior to crystallisation) on LI22. In short, by using a Bayesian approach to sample a wide variable space, we simulate multiple scenarios, some where there is a large dissolution ‘overshoot’ (i.e., a high solid/melt ratio) and some where there is only very small amounts of dissolution prior to crystallisation (low solid/melt ratio). In addition, our models allow us to assess the influence of preferential dissolution of one-phase over the others as hinted at by the reviewer here.

2. Starting melt composition: The authors do not justify the choice of averaging 3 specific samples to obtain their starting composition for the models. Why not start from the primitive SWIR MORB estimated from Coogan et al. (2004) at the Atlantis Bank, for example? The starting melt composition governs the phase saturation and might result in very different results if clinopyroxene appears before or after on the liquidus. As the authors discuss the impact of the reacted matrix on the evolution of porosity, I think that they should thoroughly discuss the impact of the starting melt composition as well (Lines 511-521).

The starting melt composition was chosen as the mean composition of the most primitive samples available for the South-West Indian Ridge (Gale et al. 2013). This is by no-means a unique solution to the starting melt composition from this area, but we do note that the fractional crystallisation models performed in this study recreate the trends observed in during magmatic differentiation in this region. As a result, we suggest that this starting melt composition is appropriate for our models and additional justification for this starting melt composition has been added to the methods section.

In addition, we have rerun all our models with a starting melt composition taken from Dick et al. (2000), based on the most primitive Fracture Zone basalt composition from the SWIR. The results of these models are presented in the supplementary information, and we find that the choice of starting composition (within reasonable bounds) has no influence on the overall results of our models.

3. Formation of porosity channels: Throughout the text, the authors mention that dissolution of plagioclase occurs at faster rates than that of olivine and clinopyroxene (Lines 165-167). Additionally, the modal composition of the considered reacted mushes is dominated by plagioclase, from troctolitic (ol:plg:cpx = ~25:70:5 vol%) to gabbroic crystal mushes (ol:plg:cpx = ~5:55:40 vol%). It thus seems fair to assume that systems where plagioclase is dissolved at rates higher than $X_{\text{React}}(\text{plg}) = 0.5-0.6$ would be the most common in nature. How do the authors reconcile the preferential assimilation of plagioclase and the large quantity of plagioclase in gabbroic crystal mushes with the decrease in porosity in reactive systems dominated by plagioclase dissolution?

In a figure that has now been moved from the Supplementary Information into the main text (Fig. 2) we show that there is a strong, and unsurprising, correlation between the fraction of a certain phase involved in the reaction processes (i.e., $X_{\text{React}}^{\text{phase}}$) and the proportion of the crystallised assemblage that is represented by that phase. In other words, if the reaction is dominated by the dissolution of plagioclase, plagioclase will likely be strongly apparent in the reprecipitated assemblage. The influence of melt-mush reaction on the mineralogy of the mush (i.e., whether or not reactions drive increasing or decreasing proportions of the different mineral phases) is now discussed in detail in the Supplementary Information (Fig. S.3). These results indicate that we generally observe a slight enrichment in the proportion of clinopyroxene that is present for the gabbroic models (consistent with natural observations; Lissenberg & MacLeod, 2016). In addition, if plagioclase-dominated reactions occur and drive the porosity of the system down, the decrease in the system porosity will partially be caused by the crystallisation of new, low An plagioclase rim zones – once again, consistent with natural observations (Sanfilippo et al. 2020).

By moving Fig. 2 from the Supplement and into the main text, adding a short discussion around the formation of enriched plagioclase and clinopyroxene crystal rims during reaction, and a more general discussion of the model results at the start of the results section (L135-144), we believe we have explained the reasoning above and demonstrate why plagioclase remains an important component (modally) in low porosity gabbroic mush systems.

Although in some cases, replenishment of clinopyroxene-undersaturated, high-temperature melt could indeed lead to increasing porosity of the system, shouldn't the porosity of the system be driven down as soon as clinopyroxene appears on the liquidus? Is there a way of creating porosity channels from more evolved, clinopyroxene-saturated melt? This has very strong implications since it preferentially links the most primitive lower crustal systems directly to the eruptible system.

The reviewer raises an important point that, if experimental observations of plagioclase dissolution occurring at rates greater than that of olivine or clinopyroxene are to be trusted, surely we should expect plagioclase-dominated reactions to be the 'norm' and therefore a decrease in the system porosity during reactive porous flow (outside of the influence from recharging melts etc.)?

We have revised the text to acknowledge this possibility, reducing the assertion that reactive flow has no overall impact on mush porosity. While we cannot confirm that plagioclase-dominated dissolution dominates melt-mush reaction in natural systems (as there are other factors such as the Ca/Na and Mg/Fe ratios of the melt phase that will likely determine which phases react with the percolating melt), we do acknowledge the experimental results that highlight this as a possibility.

We also mention throughout the revised text that, to obtain an increase in porosity, all that is required are conditions that favour clinopyroxene and/or olivine dissolution (over plagioclase). The most obvious example for this (and thus the scenario we chose to focus most of our attention on – especially given natural data for high-T clinopyroxene dissolution in gabbroic systems) is magma replenishment of clinopyroxene-undersaturated melts, but it is likely far from the only possible scenario.

4. Supplementary Figures 4-5-6: In the histograms of crystallized mass, I do not understand the significance of the peak in models that crystallized no plagioclase? Almost all models include plagioclase assimilation and the melt is initially plagioclase-saturated, it is hard to believe that so many models did not crystallize any plagioclase (Fig. 4H, 5H, 6H). I can understand that in a troctolitic world, models do not allow crystallization of cpx at high temperatures (Fig. 5I), but it is hard to believe for plagioclase, which is strongly assimilated and clearly the dominant phase on the liquidus (Fig. 6H). Same goes for all the models that crystallize 100 vol% olivine (Fig. 6G) in a high-temperature system saturated in plagioclase.

These results do indeed appear to be quite confusing at first glance, but what our results really show is that there is generally an increase in the proportion of mafic phases (cpx/ol) relative to plag during melt-mush reaction (graphs showing this have been added to the supplementary information; Fig. S.3). This is consistent with natural observations and, as a result, in models that have low amounts of assimilated plag (regardless of the wider mush composition) the system composition is dragged off the ol-cpx-plag cotectic and crystallises no plagioclase. This is now explained in more detail in the Supplementary Information as well as at the start of the results section where we provide a broader scale description of the model results (i.e., not solely focused on porosity changes).

Minor comments:

Line 11-12: This is only one of these "several questions" raised, you should rephrase this sentence.

Multiple questions are now listed here to ensure that the phrasing of the sentence makes sense.

Line 16: "under-constrained"

Changed

Lines 38-43: Compaction and reactive flow are not mutually exclusive, as the former can very well drive the latter. During compaction of any crystal mush, extraction of the interstitial melt forces percolation within a crystal matrix that is likely not in equilibrium with this migrating melt. This chemical disequilibrium can trigger dissolution-precipitation processes along the way. It is true, however, that compaction processes are hard to properly constrain and have been only recently object of detailed EBSD studies (e.g., Bertollett et al., 2019 Geology; Ferrando et al. 2021 Tectonophysics). The way the sentence is written opposes compaction to reactive flow, whereas they are two processes working together within a single environment.

We agree with this statement from the reviewer and have added an extra clause to the paragraph starting on L38 indicating that reactive flow, or melt percolation more generally, may be aided or even driven by mush compaction (i.e., that the two processes work together).

Lines 45-46: By flow, do you intend both diffuse and focused percolation?

Yes, in this study we use the term 'reactive flow' as a general descriptive term for both porous flow along grain boundaries and focus flow in reaction generated melt channels. This is now emphasised in the main text at L48.

Lines 62-64: This information already comes in handy for decades of AFC models that used ratios of assimilated over crystallized mass ($r = Ma/Mc$) of 0.95-0.99, without any constrain whether this is feasible and realistic!

Exactly! We've added to the section here to emphasise this point.

Line 94: "These include the anomalously high"

Fixed

Lines 132-135: But the dissolution of these phases are function of the temperature of the system and chemical composition of melt and crystal matrix. I think that this should be mentioned.

We have added discussion of T controls on mineral dissolution and the potential uncertainty this causes to L164 – 166 of the main text.

Lines 147-148: “interaction of olivine- and plagioclase-saturated, but clinopyroxene-undersaturated melts”

Fixed

Line 178: compaction "drives" porous flow of melts, more than requires.

Changed as suggested.

Line 188: “transfer of melt from through the”

Fixed (“from” removed)

Lines 193-194: “on a centimetre- to metre-scale in oceanic core complexes and ophiolites”

Fixed

Line 196: “the absence of a shape-preferred orientation”

Fixed

Lines 194-197: By the lack of SPO, do you mean that no extensive compaction was involved? I think that you should mention it directly, to avoid misunderstandings.

Here we mean that there was no notable deformation of the mush leading to the formation of a melt channel. An explanation of this logic has been added to the text here.

Line 390: “higher Melt Mass Ratios than in Scenario 1”

Fixed.

Figure 3: Please be more explicit in the figure caption, by dividing A and B, and explaining in detail why the melt phase decreases in proportion in B during the cpx-dominated reaction leading to an increase in porosity.

We have clarified the significance of both panels A and B in the figure caption. Panel B represents the phase proportions in the local chemical subsystem of the reaction, which is itself growing relative to the proportion of unreacted material.

Line 448: Is the “initial melt mass fraction of the mush system” to be considered as the porosity of the system?

While strictly not representing the mush porosity (due to differences in the density of the various phases) changes in the mass fraction of melt in the mush can be considered to broadly represent changes in the porosity of the mush system. This is clarified in the Methods section at this point.

Line 464: “These fractional crystallisation”

Fixed

Lines 522-533: This approach is used to capture the full variability of how the system can vary upon reaction. Is this likely that the system can lead to extensive dissolution of clinopyroxene, while plagioclase is left unreacted? A consideration of the equilibrium between the melt and the reacted crystal matrix could be used to filter the most extreme cases. I understand that the large database

created this way spans all end-members of reaction and therefore gives the whole picture of the reaction-driven chemical variability; it must then be carefully analyzed and interpreted by the user.

This is difficult point to address as there are many different factors to consider. For example, melts at different degrees of magma differentiation may be in Fe-Mg disequilibrium with clinopyroxene and olivine, but may approach Ca-Na equilibrium with plagioclase if there are a range of primary mantle melts that have different initial Ca/Na ratios (i.e., shallow vs deep melts in the melting column). As recent isotopic work has shown that substantial chemical heterogeneity remains in the lower crust of mid-ocean ridges (Lambart et al. 2019), and that enriched and depleted melts can have different phase saturation states at equivalent pressures and temperatures (Neave et al. 2019), it is not impossible to imagine that infiltration of chemically distinct melts into a magmatic mush system could drive dissolution of, dominantly, a single phase. There are also kinetic factors to consider – is dissolution of olivine likely when relatively rapid Fe-Mg diffusion may occur, alternatively Ca-Na diffusion in plagioclase is very slow so this might indicate that equilibrium is more likely to be obtained through dissolution-reprecipitation reactions?

In short, we believe that it is very difficult to constrain what would represent a ‘realistic’ or ‘most-likely’ scenario in natural systems and thus filter our models to include only these situations. Nevertheless, we note that further development of the code presented alongside this manuscript in the future could include assessment of chemical disequilibrium (e.g., by looking at the activity of various components in the liquid and solid phases) that could inform the choice of reactants. This is something that we are happy to discuss with the reviewer in more detail if they are interested.

For now, we have not made any substantial changes to the modelling or text based on this comment, with the exception of a couple of additional sentences added to L630 – 638 detailing the possible chemical heterogeneity that is likely present in mid-ocean ridge magmatic systems.

Line 536: “buoyancy-driven”

Fixed

Line 537: “high-porosity melt channel”

Fixed

Line 538: “results from buoyancy- and compaction-driven flow

Fixed

Reviewer #2 (Remarks to the Author):

Dear Authors,

I read your manuscript with interest, although I feel ill-equipped to dig very deep on the more petrological aspects of the study.

This study is concerned with the role of reactive transport in maintaining permeable flow through mafic mushes and explaining changes in mineral assemblages associated with it. To this effect a significant number (>400) thermodynamic simulations with MELTS are run varying parameters such as the initial state of the mush (temperature, assemblage, starting porosity, melt composition) to obtain equilibrated new assemblages and porosity. I do believe that the present study fills an important gap, probing if the reactions that occur along flow paths are likely to impede or facilitate further transport as they progress. I think the manuscript makes a good case that for mafic systems reactive porous flow is likely to maintain flow (no significant changes in porosity).

There are three easy to address issues I found with the manuscript and I will start from the simplest one and finish with perhaps the most tricky one.

(1) The text up to line 70+ is fairly general with respect to the types of magmatic systems considered. I did not realize before line 70 or so that this study was exclusively focused on mafic systems with mostly MOR considerations. A large fraction of the intro references papers that are written with more silicic and continental crust oriented systems. I would encourage the authors to make the text more specific from the start (even in the title) and reduce the intro to the discussion of mafic systems, as I have doubts that the present study provides significant evidences for the consequences of reactive flow in silicic mushes.

In our initial submission we had attempted to keep the introduction relatively broad and generalised as the methods developed in this study could easily be expanded to investigate the influence of reactive flow in dioritic to granitic mush systems. However, we acknowledge that most of our results and discussion focus on the influence of reactive flow in mafic mush systems and we have made several changes to the abstract and introduction to highlight this fact. For example, on L56 we now specify that AFC has specifically been used to assess chemical enrichment in troctolitic, gabbroic and wehrlitic mush systems. In addition, we state that we examine the influence of reactive flow on the porosity of “troctolitic to gabbroic mush systems, relevant to mid-ocean ridge magmatism” on L63-64.

(2) The intro was a bit disappointing to read and, I am sorry to say, felt a bit lazy. First the lack of focus (see previous point) and second the choice of references used in the text is frustrating. About the latter, the authors rightfully point that the mush paradigm has been developed about 2 decades ago, yet all the references in the first paragraphs are recent and mostly from a single, yet prominent, group in a much larger community. Please cite the original studies that presented the mush paradigm as well as the recent studies that banked on it (see papers from Wes Hildreth, Olivier Bachmann and other papers by Bruce Marsh). Also update the references (line 32-33) about the geophysical support for the lack of melt-rich reservoirs with recent papers that explain that the limitations of seismic tomography are significant and their absence in the geophysical record not preclude their existence (see for example Rashtbehesh et al, JGR, 2020).

We have added a significant number of references, as well as a short summary of the identification of mush reservoirs in first MOR settings and then at other volcanic systems worldwide, to the introduction of the manuscript. These changes correctly reference and acknowledge the original

work on mush reservoirs, including some of the suggestions provided by the reviewer, and also draw more attention to the work done on mid-ocean ridge magmatic systems, highlighting the reviewers first comment as well.

(3) It seems that most (all?) MELTS simulations were run under fixed enthalpy conditions. If this is correct, then the authors should clearly justify this choice and perhaps discuss some of the implications.

In all the models shown in the main text we have assumed that the enthalpy of the mush system (i.e., both reacted and unreacted solid components in addition to the melt phase) remains constant through melt-mush reaction. This assumption was taken to allow our models to focus onto the influence of key mineralogical, chemical and thermal parameters for melt-mush reaction (and reactive flow) on a relatively small scale (i.e., a narrow layer within a mush system likely a few cm to dm wide). To summarise, our models represent a scenario where there is no heat transfer into or out of that local system (so that the temperature of the mush is controlled by the energy exchange during the reaction).

Models of reactive flow with a greater focus on the physical transfer of melt and solid throughout the system, accounting for the differential transport of crystal and solid phases within the mush system (for example, the model of Jackson et al., 2018), apply similar logic/assumptions to each 'layer' or z-interval within their model (i.e., find the new 'equilibrium' state for a specified enthalpy/heat energy) with additional calculations then required to account for heat transfer through the movement of melt and crystal phases and/or conductive heat transfer.

Unfortunately, these models contain only very simplified chemical parameterisations, leaving important uncertainties regarding the influence of mush mineralogy/attainment of partial equilibrium across a mush horizon. Therefore, in this study we aimed to address these issues by performing a 'deep-dive' investigation of the chemical systematics of melt-mush reaction, accounting for a heterogenous solid component and only partial dissolution of the initial mush framework. Our thermodynamics-based model approach does this, and we consider our models as representative of the processes operating within a single location/level (and time) of a mush system (i.e., a single component of one of these larger-scale reactive flow models). Our results demonstrate the importance of considering the complexities of chemically and mineralogically heterogeneous mush systems (that is, the 'solid' in reactive flow models cannot be treated as a single, chemically homogeneous component) and our models could potentially be combined with larger-scale models of melt and crystal transport in the future.

We have attempted to emphasise how our models are relevant to the processes operating in natural systems throughout the main text and highlight all of the issue/topics raised in this response.

My understanding (but I could be convinced otherwise) is that because of the equilibrium assumption built in these thermodynamic models, it is assumed that the final assemblage is fully reacted/equilibrated. How good of an assumption is that? Is it possible to consider that kinetically limited conditions can lead to different evolution paths and perhaps more porosity change than the comparison between initial and fully reacted assemblages (given that the small change in porosity is mostly attributed by the dissolution of certain minerals being balance by precipitation of other phases)?

The reviewer is correct that the thermodynamic nature of these models means that we are required to assume that the final assemblage of the local melt + reacted solid chemical system is fully equilibrated. However, owing to our method of sampling only a fraction of the initial solid to be "reacted" (i.e., incorporated into the same chemical system as the melt phase via dissolution and diffusion-controlled processes) we do not require that the entire mush system is in chemical

equilibrium at the end of each model step (we do, however, require thermal equilibrium across the mush).

Kinetic factors could, theoretically, stop crystallisation of a particular phase (i.e., if the energy barrier for nucleation is too large), which would mean that the final assemblage may not represent anything approaching true chemical equilibrium. However, as the recrystallisation step likely only includes crystallisation on pre-existing grains (rather than nucleation of new phases) the influence of these kinetic factors may be minimised.

We have clarified how our approach calculates the local equilibrium of the melt + reacted solid chemical subsystem, and thus only represents partial chemical equilibrium across the mush itself. Additionally, we have clarified in the main text and methods how our Bayesian approach, in combination with this partial equilibrium approach, may help us assess the possible influence of kinetic factors (e.g., variations in dissolution rates).

Under several instances, the text refers to melt transport, compaction processes or even rapid changes (line 215). These are misleading because the thermodynamic results presented here do not have lengthscales or timescales attached to them. I would be more careful with these statements. I would like to encourage the authors to discuss equilibrium and the absence of kinetics a bit more, especially in the discussion section. There is only so much one can do with MELTS type simulations.

The reviewer is correct in that the thermodynamic results cannot offer us any insights into the timescales or lengthscales of the different processes discussed. Where timescales and lengthscales are discussed in the main text, this is done with reference to generalised processes that are known to occur in magmatic systems, potentially supported by some back of the envelope calculations (e.g., porous and channelised fluxes, L239 and Fig. 5). In no case did we intend to suggest that our MELTS models could provide information about the timescale or lengthscale of processes in magmatic systems and we have carefully read through (and modified the text where necessary) to avoid any potential confusion regarding this topic.

Reviewers' Comments:

Reviewer #1:

Remarks to the Author:

A revised version of the manuscript "Porosity evolution of mafic crystal mush during reactive flow" has now been resubmitted by Matthew Gleeson and co-workers to Nature Communications. Overall, the authors addressed properly the reviewer's comments, and I thank them for the effort in adapting the initial version of the manuscript. The revisions improve the clarity of the models and results; I only have few very minor comments, listed hereafter, before the manuscript can be accepted for publication.

I also checked the replies to the comments made by reviewer #2 and provide below a short comment on each of the responses. I reckon that the manuscript was modified properly, according to the reviewer's comments.

As I previously stated in the first round of revision, this contribution will be a welcome addition to the field of magmatic petrology dealing with crustal processes of reactive flow.

Additional minor comments to be addressed:

Lines 49-51: "Existing numerical models ... within crystal mush systems" - This sentence feels like the authors are describing numerical models of the physical movement of melts. I reckon that it should be specified that they here mainly address the chemical aspect of reactive porous flow.

Lines 80-81: This seems a bit reductive, because these models can constrain more than the build-up to volcanic eruptions. It allows to better constrain melt migration within the crust as a whole. This does include the volcanic system but it is a part of a bigger melt migration system.

Lines 196-279: A large number of the referenced bibliography in results and discussions are self-citations. I think that they should be avoided not to lose in credibility, especially in the context of reactive porous flow, which is extensively reported in literature.

Reviewer #2

Responses to reviewer's comments:

- 1) "The text up to line 70+ ..." - The authors did modify the title, abstract and introduction according to the reviewer's comment. I reckon that this version of the introduction is more to the point regarding the system that is studied. However, I appreciate that the authors kept the general introduction as well, since these results could easily be extended to silicic systems. The main feature that is studied is the reactive porous flow process, which occurs in all magmatic settings.
- 2) "The intro was a bit ..." - The authors made the necessary corrections to answer the reviewer's comment.
- 3) "It seems that most (all?) MELTS ..." - The authors answer thoroughly the reviewer's comment and included within the manuscript the related explanations. I reckon that the author's reasoning is correct and in order to properly document the effect of melt-rock interactions on the percolated rock and porosity of the system, their approach is reasonable.
- 4) "My understanding (but I could be convinced otherwise) ..." - The authors answered the reviewer's comment extensively and modified the manuscript accordingly.
- 5) "Under several instances, the text ..." - The reviewer's comment has been included within the revised version of the manuscript.

Reviewer #1 (Remarks to the Author):

A revised version of the manuscript “Porosity evolution of mafic crystal mush during reactive flow” has now been resubmitted by Matthew Gleeson and co-workers to Nature Communications. Overall, the authors addressed properly the reviewer’s comments, and I thank them for the effort in adapting the initial version of the manuscript. The revisions improve the clarity of the models and results; I only have few very minor comments, listed hereafter, before the manuscript can be accepted for publication.

I also checked the replies to the comments made by reviewer #2 and provide below a short comment on each of the responses. I reckon that the manuscript was modified properly, according to the reviewer’s comments.

As I previously stated in the first round of revision, this contribution will be a welcome addition to the field of magmatic petrology dealing with crustal processes of reactive flow.

Additional minor comments to be addressed:

Lines 49-51: “Existing numerical models ... within crystal mush systems” - This sentence feels like the authors are describing numerical models of the physical movement of melts. I reckon that it should be specified that they here mainly address the chemical aspect of reactive porous flow.

While we aim to address the chemical aspects/systematics of reactive flow/melt-mush reaction, we are here referring to existing models that have incorporated both physical and chemical considerations into their models of reactive melt transport. This is now outlined more clearly in the text at the appropriate point.

Lines 80-81: This seems a bit reductive, because these models can constrain more than the build-up to volcanic eruptions. It allows to better constrain melt migration within the crust as a whole. This does include the volcanic system but it is a part of a bigger melt migration system.

Thank you for this suggestion. We completely agree and have modified the text accordingly to outline the wider implications of our research for understanding melt transport and crustal magma system construction as well as the dynamics prior to volcanic eruptions.

Lines 196-279: A large number of the referenced bibliography in results and discussions are self-citations. I think that they should be avoided not to lose in credibility, especially in the context of reactive porous flow, which is extensively reported in literature.

We have added ~10 new citations to the manuscript, several of which are recent publications from late 2022 or early 2023 to demonstrate a wider appreciation for the work on reactive flow that has previously been completed.

Remarks on Reviewer #2 comments:

1) “The text up to line 70+ ...” - The authors did modify the title, abstract and introduction according to the reviewer's comment. I reckon that this version of the introduction is more to the point regarding the system that is studied. However, I appreciate that the authors kept the general introduction as well, since these results could easily be extended to silicic systems. The main feature that is studied is the reactive porous flow process, which occurs in all magmatic settings.

2) “The intro was a bit ...” - The authors made the necessary corrections to answer the reviewer's comment.

3) “It seems that most (all?) MELTS ...” - The authors answer thoroughly the reviewer's comment and included within the manuscript the related explanations. I reckon that the author's reasoning is correct and in order to properly document the effect of melt-rock interactions on the percolated rock

and porosity of the system, their approach is reasonable.

4) “My understanding (but I could be convinced otherwise) ...” - The authors answered the reviewer's comment extensively and modified the manuscript accordingly.

5) “Under several instances, the text ...” - The reviewer's comment has been included within the revised version of the manuscript.

We would like to thank the reviewer for taking the time to check our responses to Reviewer 2, and for the positive comments regarding our response/edits.